# Experiences of women undergoing assisted reproductive technology in Ghana: A qualitative analysis of their experiences

**Judith A. Anaman-Torgbor**[1], **Justice Wiston Amstrong Jonathan**[2]*, **Lily Asare**[3], **Bernice Osarfo**[3], **Rita Attivor**[3], **Afia Bonsu**[3], **Elizabeth A. E. Fialor**[3], **Elvis E. Tarkang**[4]

**1** Department of Public Health Nursing, School of Nursing and Midwifery, University of Health and Allied Sciences, Ho, Volta Region, Ghana, **2** Department of Basic Sciences, School of Basic and Biomedical Sciences, University of Health and Allied Sciences, Ho, Volta Region, Ghana, **3** Department of Nursing, School of Nursing and Midwifery, University of Health and Allied Sciences, Ho, Volta Region, Ghana, **4** Department of Population and Behavioural Sciences, School of Public Health, University of Health and Allied Sciences, Hohoe, Volta Region, Ghana

* jjonathan@uhas.edu.gh

**Data Availability Statement:** All relevant data are within the manuscript and Supporting Information files.

## Abstract

### Objective

The study aimed to explore the experiences of women undergoing Assisted Reproductive Technologies namely; Invitro Fertilization and Intracytoplasmic Sperm Injection at the Finney Hospital and Fertility Centre, New Bortianor, Ghana.

### Method

A qualitative research design was employed to analyse and describe the experiences of the women seeking Assisted Reproductive Technologies. A total of 32 women were invited to take part in the interview, 15 of them accepted the invitation. However, saturation was reached before all interviews had been complete.

### Results

Three themes emerged from the study: the women's experiences, challenges and the roles and contributions of *significant others*. The women were anxious, stressed-up, exhausted and financially burdened. Spouses and health professionals played significant roles by providing social, emotional and financial support for these women. Significant others such as spouses and close relatives were supportive and provided encouragement to the women.

### Conclusion

The experiences of women undergoing Assisted Reproductive Technologies are multi-dimensional. Thus, psychosocial interventions as part of ART services with health insurance cover may be client-centered and more appropriate for these group of women.

**Funding:** Authors received no specific funding for this work.

**Competing interests:** The authors have declared that no competing interest exist.

## Introduction

Child bearing is an importance milestone in the life of most couples, especially in Africa, of which Ghana is no exception. The widely accepted belief is that the life of humans reaches completeness through the birth of a child. This is because the birth of a child is a fulfilment of an individual's need for reproduction [1]. Thus, when couples live together and are engaged in regular unprotected sexual intercourse without achieving the goal of pregnancy, they are often diagnosed with infertility. This excludes transgender individuals, gays and lesbians couples. A voluntary decision to be childless is an uncommon practice in Ghanaian traditional setting because, the ultimate expectation of every marriage is to reproduce. Thus, infertility affects the stability of marriages in Ghana [2]. Primary infertility refers to couples who have never been able to conceive in their lifetime, while secondary infertility refers to couples who have either carried a pregnancy to a full term or have had a miscarriage in the past and are unable to conceive again. Infertility cuts across genders, cultures, races and socioeconomic classes. A review of previous studies from 1990 to 2006 reported prevalence rates in a 12-month range as 3.5% to 16.7% in developed nations and 6.9% to 9.3% in less developed countries [3]. In Ghana, it is estimated that the prevalence of infertility stands at 15% among Ghanaian women and 15.8% among men [4].

In recent times, orthodox medicine has helped to keep the hopes of infertile couples alive. This is significant because infertile couples have the opportunity to have children through the use of Assisted Reproductive Technologies (ARTs). Likewise, same sex couples (lesbians and gays) can also achieve the joy of having children in marriage [5]. ARTs are medical interventions that are used to help childless couples to have their biological babies. In vitro fertilization (IVF), Gamete Intra Fallopian Transfer (GIFT), Pronuclear Stage Tubal Transfer (PROST), Tubal Embryo Transfer (TET), and Zygote Intra Fallopian Transfer (ZIFT) are some examples of such interventions [6]. IVF has contributed to population growth more than expected, reaching about 3.5% of the world population [7]. These figures are still projected to expand in the near future. It is not surprising that locally and internationally, the demand for ART services is growing exponentially. ART services may be skewed towards high-income level countries [7] whereas the sub-Saharan region is characterized by fewer IVF services [8]. Nonetheless, South Africa, Nigeria and Ghana in sub-Saharan Africa have experienced comparative regional success stories [8]. After the first successful IVF in 1995 in Ghana, several other clinics in the country commenced ARTs services to women with some degree of success.

Although ARTs have brought some measure of hope to infertile and childless couples, it has also brought untold economic, social, moral, legal and emotional burden onto women. Moral issues and legitimacy of children born with the help of ART involving a donor remains unresolved. Women would have to wait for several weeks without knowing whether the ART process was going to be successful or not. ARTs are rejected by many as intrinsically morally unacceptable because fertilization takes place outside the body. The process of fertilization outside body is believe, separates reproduction from sexual intercourse or does not associate reproduction with sexual intercourse. This is considered as morally unacceptable by some people. Some women expressed that going through the ART process make them feel as though they had lost their sense of control. This is because they had to wait not knowing what the outcome of the process was going to be. The desire for couples to have children after marriage is so intense that when pregnancy or childbirth continues to elude couples, it generates fear and anxiety with women disproportionately suffering the consequence the most.

Considering the fact that in many traditional Ghanaian settings, the natural means of having children is more preferable to society, raising pertinent questions such as:

❖. What are the experiences and challenges women go through accessing ART services?

❖. What is the role of significant others in the life of the women seeking ART services?

These are the nagging questions this research is seeking to address by exploring the experiences of women undergoing ART at the Finney Hospital and Fertility Centre, New Bortianor in Ghana. **Table 1** below presents the statement of significance.

## Materials and methods

### Design

A qualitative research design with content analysis was employed to describe the experiences of the women seeking Assisted Reproductive Technologies at the Finney Hospital and Fertility Centre in Ghana. Qualitative research design was used because it is the most appropriate to explore and understand the experiences of the women seeking Assisted Reproductive Technology services.

### Study population and sampling strategy

The population for the study were mainly women Assisted Reproductive Technologies services and women who were referred from other health facilities or who visited the Finney Hospital and Fertility Centre seeking Assisted Reproductive Technologies services. Purposive sampling technique was used [9]. Purposeful sampling is widely used in qualitative research for the identification and selection of information-rich cases related to the phenomenon of interest. In the study, the phenomenon of interest was experiences of women undergoing Assisted Reproductive Technologies thus women with the appropriate characteristics were identified and invited. Data collection spanned from November 2019 to January 2020. During this period, the research team was at the facility, identified and approached potential participants individually. Those women who met the inclusion criteria were invited and provided with Participants Information and Consent Form (PICF). The women were informed that participation was voluntarily. Those who were willing to take part in the study joined after written consents were obtained. Participants were then taken individually to a room at the facility and interviewed. Generation of a large and representative sample is not the primary focus of qualitative research [10]; rather, the purpose is to explore meanings and experiences to add another layer to the existing scientific evidence. A total of 32 women were invited to take part in the interview, 15 of them accepted the invitation. However, ssaturation was reached before all interviews had been complete [10], i.e. no new themes emerged from the last few interviews.

**Table 1. Statement of significance.**

| | |
|---|---|
| **Problem or Issue** | Although ARTs have brought some measure of hope to childless couples, it has also brought untold challenges. |
| **What is already known** | Modern orthodox medicine has helped to keep the hopes of infertile couples alive by having children through the use of Assisted Reproductive Technologies (ARTs). By extension, same sex couples (lesbians and gays) can also achieve the joy of having children in marriage. ART has contributed to population growth more than expected, reaching about 3.5% of the world's population but very little is known about the experiences of women who seek this intervention in the Ghanaian setting. |
| **What this paper adds** | Evidence that women seeking ART services in Ghana experience anxiety, stress, frustration and are burdened with the challenge of funding the treatment. Thus, the need for a psychosocial intervention as part of ART services and insurance cover for ART may be appropriate. |

## Inclusion and exclusion criteria

Women were included in this study if they were undergoing fertility treatment at the Finney Hospital and Fertility Centre using any of the assisted reproductive technologies, were in their first cycle or a repeated cycle and were willing to take part in the study. Women were excluded it they did not experience any of assisted reproductive technologies.

## Data collection tool and process

An interview guide was developed based on the study objective and after a review of the relevant literature. The interview guide was in two sections: the first section was concerned with the socio-demographic characteristics of the participants while the second section was to elicit information on the experiences of the women undergoing ARTs, their challenges in accessing treatments and the role of significant others in the ART process. All the participants could speak and understand English therefore all the interviews were conducted in English by the Principal Investigator (PI) who has a strong background in qualitative research. The PI is nurse with Postgraduate qualifications in Public Health. She completed research methodology coursework while undergoing her postgraduate program. The knowledge acquired from this training has been applied to this study. After consenting to participate in the study, interviews were scheduled based on dates and times suitable to the participants. The interviews were audio-recorded and field-notes were also taken. Each interview lasted between 45 minutes and one hour. Probing questions was used to ensure in-depth information of the subject matter.

## The position of the researcher

It's known that a researcher's background and position may influence data collection and interpretation [11] and indicating how researcher's beliefs and values came into play during the research process is a common strategy to negotiate certain knowledge claims [12]. Therefore, this study endeavoured to capture the views and expressions of the study participants in all their original complexity and depth, without distorting the meaning. All the stages of the research process; data collection, interpretation and reporting of results were shaped by her personal and social characteristics and the researcher's understanding of the research objectives. However, the researcher focused using an emic approach in the analysis and presented diverse perspectives and maximized the use of direct quotations from the participants.

## Data analysis

The data were manually analysed using content analysis to identify patterns across the data sets [13, 14]. Content analysis was considered for this study because of its flexibility; it is applicable to a data-driven or theory-driven analysis. The analysis started with immersion into the data, involving verbatim transcriptions of all the audio recordings by the Principal Investigator and verified independently by the research team members. The team read the interview scripts multiple times in order to accurately capture the accounts of participants' experiences. The transcripts and the field-notes taken during the interviews were reflected upon and coded. Key themes were identified from the different transcripts and displayed in matrices to enable systematic examination of similarities and patterns among various responses [14].

## Ethical considerations

A number of activities were employed to establish the credibility of the study. Experts in reproductive health and qualitative research contributed to this study. The data analysis endeavoured to capture and retain the women's expressions in their original depth without any

distortion. The analysis and interpretation of the data were reviewed by all study team members. Ethical clearance was obtained from the University of Health and Allied Sciences Research Ethics Committee (UHAS-REC. A.1 [47] 19–20). The purpose of the study and procedures were explained to the women with the aid of participants information and informed consent guide. Confidentiality was ensured and data collected were sealed and stored under key and lock. All the research team members verified the transcript and field-notes used to corroborate the women's accounts. Some of the women were followed-up on for further clarifications and confirmation of their accounts.

## Results

### Participants characteristics

Fifteen women provided accounts of their individual experiences of ART process. All the women were above 34 years, thirteen were married, eleven were leaving with their spouses and only two had ever conceived. One woman had a child about 19 years ago. Ten were self-employed, five had a Master's Degree, while the remaining ten were educated below a Bachelor's Degree level. **Table 2** presents a summary of the participants characteristics.

### Themes

Three themes emerged from the data analysis and these are presented in Table 3.

**Theme 1. Experiences of women undergoing ART.**   The findings from the study revealed several reasons behind women seeking ART. Some of the reasons expressed by these women were: to escape social stigma of childlessness, the desire to satisfy their spouses and for some of these women, to avoid the displeasure of their in-laws. However, for majority of the women, the strong desire to become pregnant was what motivated them to seek for ART. Most of them have been married for years and had made several attempts to conceive naturally and they were not successful. All the participants were concerned about their current ages because they are getting older without children and therefore they needed help. One of the participants stated that:

**Table 2. Participant characteristics.**

| Participants | Age (Years) | Occupation | Education | Marital status | Living with spouse | Conceived before |
|---|---|---|---|---|---|---|
| Participant 1 | 42 | Trader | SHS graduate | Married | Yes | Yes, has one child. |
| Participant 2 | 36 | Seamstress | Voc. Training | Divorced | No | No |
| Participant 3 | 44 | Hairdresser | JHS graduate | Married | Yes | No. |
| Participant 4 | 37 | Administrator | MBA | Married | Yes | No |
| Participant 5 | 43 | Educationist | MSc Degree | Married | Yes | No |
| Participant 6 | 34 | Teacher | Diploma | Married | No | Conceived twice but lost all |
| Participant 7 | 37 | General manager | MBA | Single | No | No |
| Participant 8 | 35 | Business woman | Diploma | Married | Yes | No |
| Participant 9 | 40 | Trader | SHS graduate | Married | Yes | No |
| Participant 10 | 34 | Banker | Master's degree | Married | Yes | No |
| Participant 11 | 38 | Trader | SHS graduate | Married | Yes | No |
| Participant 12 | 40 | Trader | SHS graduate | Married | Yes | No |
| Participant 13 | 40 | Trader | SHS graduate | Married | Yes | No |
| Participant 14 | 35 | Teacher | Diploma | Married | No | No |
| Participant 15 | 34 | Banker | MBA | Married | Yes | No |

**Table 3. Themes.**

| THEMES | HEADINGS |
| --- | --- |
| Theme 1 | Experiences of women undergoing Assisted Reproductive Technology |
| Theme 2 | Challenges women face as they seek Assisted Reproductive Technology |
| Theme 3 | Role of significant others in the ART procedure. |

*"I am concern about my age. My husband and I, we both felt we were growing old and we needed children. We read somewhere that it is better to give birth early and we decided to have a child early but things became difficult. . ." (Participant 4)*

Another participant also expressed that:

*"After 5 years of marriage, I felt age was catching up with me so I needed some medical help if I wanted to conceive so I decided to come here to look for help for pregnancy" (Participant 10)*

A majority of the women interviewed experienced anxiety and emotional distress. They expressed that they were anxious going through the treatment process because of the uncertainty about the treatment outcome:

*"I have tried for 6 times and at the first attempt I thought I will just succeed. . . . . . I thought of this option as a solution to my problem so that people will not mock at me again but it didn't work the first time. . . . . . . . ." (Participant 15).*

*"I came to the hospital because of a child so I was happy at the beginning but later I got very anxious because I got to realized that it's a 50% chance and therefore I had to pray hard. Not knowing the outcome of the treatment makes me very anxious" (Participant 3).*

The women also indicated that the ART process was painful, exhausting, time consuming and tortuous and at a point they were considering giving up. Majority of them complained about the number of laboratory investigations, scans and medications involved:

*"I was a bit disturbed about the scans because the frequency of doing scan was too much. I think there should be a way to reduce the scans. Because I feel very uncomfortable with the scans and I was not enjoying it. The egg retrieval part is also very painful. I threw up a lot after the procedure" (Participant 14)*

Only few of the women indicated that going through the ART process has contributed positively to their sexual life. According to these women their sexual lives weren't good and pleasant, however, the ART process has improved their sexual life significantly. Of the 11 women who were living with their spouses, only two shared that the ART process has positively impacted their marriage lives. According to them the ART also gave their spouses some hope of having babies of their own and this they believe rejuvenated their marriages:

*"It has brought happiness to my life. Because it's been long since I gave birth and it feels as if I have never given birth before, because I have no children with my current husband. But now my husband and I are very happy." (Participant 1).*

As the women shared their experiences the study further explored and identified challenges in seeking ART services. This may help the ART service providers to consider appropriate strategies to remove these challenges.

**Theme 2. Challenges women faces as they seek ART.** All the women irrespective of their occupation or educational background complained of financial challenges. According to these women, the ART was expensive and they were able to afford the treatment because they sacrificed other needs in order to saved money towards the treatment; others obtain loans from the banks and others also said they were supported by their spouses:

*"I have been saving for about 7 years. . . . . . .. . .a lot of sacrifices, we sold our car in July just before we started the IVF. We had to sell it. I could have built a house with my money. I got the opportunities to travel to the US and while in the US I never went shopping because I had to save the money for the ART. . . . . . . . . . ." (Participant 2)*

*"I had to safe for about 6 months. I Had to go for a loan to be approved. So, I used my shop as collateral to get the loan",*

Some of the participants indicated that they were even still paying for debts incurred as a result of the ART:

*"I took a loan from the bank and I am now paying it back about six months now and for a couple of more months to come." (Participant 6)*

Due to the financial burden, one participant stated that her husband tried to convince her to give up so they could save the money being spent on ART for other things:

*"My husband thought of us stopping because of the money but for me I was interested in getting a child so I never got exhausted. Because money is nothing but children are important" (Participant 1)*

In addition to the financial burden experienced, all the women described going through the ART procedures as tortuous, stressful, time consuming and very exhausting. The women expressed the following:

*" It has been quite tedious travelling to and from the hospital, the injections, the disappointments at the negative tests. Hmmmmm! She exclaimed. It has not been easy. Severally, I had thought about giving up and adopting or just living my life" (Participant 10)*

*"The injections are on daily basis and we were coming from home for the injection, and it was very stressful. Sometimes we come late and all that, so I felt like giving up" (participant 4).*

There were challenges confronting the women going through the ART period and financial burden was largely experienced by all the women interviewed. Some of the women funded for the therapy by themselves whereas others were supported by their spouses. While some of the women were being supported by their spouses, the study observed that one woman's husband wanted her to stopped the therapy and save the money. It was therefore important to explore the role of significant others during ART.

**Theme 3. Significant others and their Role in the ART procedure.** The study also sought to identify the women's significant others and their roles during the ART procedures. When the participants were asked about family and friends who were involve in the ART

process they were going through, it was observed that these women preferred to keep their treatments a secret between them and their spouses largely to avoid the displeasure of people. Majority of support the women received during the ART process came from their spouses, including financial support. One participant indicated that:

"*In addition to the huge amount of money that my husband is spending for this treatment, I also got emotional support from my him. My husband has been very supportive*" (Respondent 9).

A few of the women involved other family members such as siblings and mothers during the ART process. According to them, it is because of the support they were receiving from these individuals. Among those who informed their mothers, one stated:

"*My mom has been very encouraging; she does not understand the process but whenever I talk to her, she asks if it has worked yet. She encouraged me to keep on praying and I should not stop*" (Participant 12)

Although some of the participants were supported by their family members, some of the women were of the opinion that their family members showed open disapproval of the ART and tended to discourage them from undergoing the procedures. For these group of women, they would prefer to involve their church pastors instead. One woman for instance, said she involved her church pastors so that they would support her with prayers:

"*My two pastors were aware of my situation but nobody else knew because they would never support me through such thing so I kept all this to myself. Furthermore, some people will end up discouraging you but the pastors always pray for me. *" (Participant 2).

Majority of the women also mentioned that they received support and encouraging words from health workers at the facility. The doctors and nurses provided encouragement and showed positive attitude toward them. These kind gestures helped majority of them from giving up. One woman expressed the following:

"*The nurses have been very helpful too. The doctors try to explain the process as much as possible. It has been good so far.*" (Participant 8)

"*My nurse, especially, was very encouraging and supportive throughout. The doctors too made me feel comfortable and relaxed at all times. Even at odd hours*" (participant 7)

## Discussions

The study found that women undergoing ART have personal experiences resulting from their expectations and the procedures involved in the treatment process. Although a majority of the women felt very happy at the first instance of knowing that they have another chance of getting pregnant to have a child, they later became very anxious. This is because after the commencement of the ART procedures these women were confronted with uncertain outcome of the ART. Their anxieties heightened as some experienced fear, treatment failures and burdened financially. This finding is consistent with a previous study in Iran by [15] that reported fear of the unknown among women who were not confident about their treatment outcome and this was a great source of distress. Despite the aforementioned challenges associated with ART,

infertile women go for the procedure in their quest to have children, the problem is heightened by the fact that they have to save for long periods, take bank loans, or gain support from relatives in order to cover for treatment cost. Even more worrying is that there is no certainty about the success of the procedure. An indication that the ART procedure can sometimes fail. The women in the study indicated that the ART procedure was tortuous, stressful, time consuming and exhausting. This finding is consistent with other studies in Iran and China that reported that women, including their families, go through physical and emotional pain, uncertainty, low self-esteem, stress, distress and frustration when undergoing ART [15–17]. The findings in this current study suggest that women seeking ART services in Ghana experience anxiety, stress, frustration and are burdened with the challenge of funding the treatment. Thus, the need for a psychosocial intervention as part of ART services and insurance cover for the ART may be appropriate. These women clearly need support however, it is also not whether there are adequate support systems for women accessing ART services. It is therefore necessary that further studies investigating ways to lessen the burden for women undergoing ART are conducted so that appropriate interventions may be developed for these women.

Whether these women had failed attempts or not, the desires to overcome the related challenges and carry pregnancy to term and deliver their children successfully were the ultimate goal. They strove to get pregnancy and once that is achieved, they were satisfied, fulfilled and all the struggle and pains are forgotten. According to the women, they were very happy getting pregnant and the feeling of having a child despite the difficulties was very great for them and their spouses. Similar finding was reported by Dornelles, MacCallum [18], who revealed that women in Brazil who initially experienced failed attempts of ART and later achieved success with ARTs regarded their pregnancies as a reward or as a compensation for the difficulties they had experienced. Such was the sentiments echoed by the nineteen expectant first-time mothers from Brazil who conceived through Assisted Reproductive Technology treatment [18].

The study findings revealed two main challenges facing these women and this included the time-consuming aspect of the procedure and the financial burden associated with it. Funding ART for majority of the women required several years of saving and making sacrifices such as using income intended for such wealth as cars and buildings for the ART procedure. These findings are consistent with the findings of Ranjbar, Behboodi-Moghadam [15] which revealed that with respect to the financial difficulties encountered, some participants highlighted that raising funds for the treatment came at great cost to them. Some had to take bank loans for the ART as insurance services did not cover their treatment cost. For some, it meant reorganizing their priorities in life; sacrificing purchasing a house or a car for having children [15]. It is not clear the impact of these challenges on the treatment outcome. Nonetheless, these challenges can be lessened if ART is considered and included in health insurance schemes for these women who are already troubled, stressed and uncertain about the treatment outcome.

Other individuals play critical roles in the success of the ART procedure apart from the women undergoing the procedure. In this study, it was revealed that the spouses of the women played significant roles by providing financial and emotional support to the women during the process. This finding correlates with a study by Pedro and Mwaba [19] who found that, ARTs have experienced a surge in its usage as a means of having children and the reason for the increased patronage, according to the author, was that the infertile couples had decided not to be discouraged by the negative attitudes of their relatives and society towards ART.

Health professionals were also supportive during the ART process. The study revealed that the women undergoing ART were encouraged by positive attitudes shown by doctors and nurses. The doctors and nurses were empathetic and showed positive attitudes toward these women throughout the ART process.

## Conclusions

There is a solid evidence that women seeking ART services in Ghana experience anxiety, stress, frustration and are burdened with the challenge of funding the treatment. These women had to endure these challenges to escape the stigma of childlessness. It is not clear how these challenges impact upon the treatment outcome and no medical intervention can change the desires women feel towards child-bearing. However, the burden of these women can be lessened if their challenges are viewed in a broader context. The social, emotional and financial support from significant others such as spouses, relatives and healthcare workers were felt and appreciated by the women. These kinds of support may not be adequate, reliable and consistent. A psychosocial intervention as part of ART services may be more appropriate and client-centered for women undergoing ART procedure.

There is no doubt that ARTs have brought some measure of hope to infertile and childless couples and also the opportunity to escape social stigma of childlessness. As the number of ART clinics in Ghana continue to grow the issues concerning ART largely remain unexplored, as such ART has not been given much policy attention in Ghana. The position of the government and churches regarding ART for instance are needed. Considering that very few studies have been conducted in Ghana regarding ARTs, the finding of this current study may contribute to the consideration of ART to be included in the health insurance schemes so that these women can benefit from health insurance cover.

## Supporting information

**S1 Appendix.**
(DOCX)

## Acknowledgments

Many thanks to all the women who participated in this study We also thank management and staff of Finney Hospital and Fertility Centre for their cooperation and support. Finally, we are grateful to University of Health and Allied Sciences Ethics Review Committee for reviewing and approving the protocol for this study.

## Author Contributions

**Conceptualization:** Judith A. Anaman-Torgbor, Justice Wiston Amstrong Jonathan, Lily Asare, Bernice Osarfo, Rita Attivor, Afia Bonsu, Elizabeth A. E. Fialor, Elvis E. Tarkang.

**Data curation:** Judith A. Anaman-Torgbor, Justice Wiston Amstrong Jonathan, Lily Asare, Bernice Osarfo, Rita Attivor, Afia Bonsu, Elizabeth A. E. Fialor.

**Formal analysis:** Judith A. Anaman-Torgbor, Justice Wiston Amstrong Jonathan, Lily Asare, Bernice Osarfo, Rita Attivor, Elvis E. Tarkang.

**Funding acquisition:** Judith A. Anaman-Torgbor, Lily Asare, Bernice Osarfo, Rita Attivor, Afia Bonsu, Elizabeth A. E. Fialor.

**Investigation:** Judith A. Anaman-Torgbor, Justice Wiston Amstrong Jonathan, Lily Asare, Bernice Osarfo, Rita Attivor, Afia Bonsu, Elizabeth A. E. Fialor, Elvis E. Tarkang.

**Methodology:** Judith A. Anaman-Torgbor, Justice Wiston Amstrong Jonathan, Lily Asare, Bernice Osarfo, Afia Bonsu, Elizabeth A. E. Fialor, Elvis E. Tarkang.

**Project administration:** Judith A. Anaman-Torgbor, Justice Wiston Amstrong Jonathan, Lily Asare, Bernice Osarfo, Rita Attivor, Afia Bonsu, Elizabeth A. E. Fialor.

**Resources:** Judith A. Anaman-Torgbor, Justice Wiston Amstrong Jonathan, Lily Asare, Rita Attivor, Afia Bonsu, Elizabeth A. E. Fialor.

**Software:** Judith A. Anaman-Torgbor.

**Supervision:** Judith A. Anaman-Torgbor, Justice Wiston Amstrong Jonathan.

**Validation:** Judith A. Anaman-Torgbor, Justice Wiston Amstrong Jonathan, Lily Asare, Bernice Osarfo, Rita Attivor, Elvis E. Tarkang.

**Visualization:** Judith A. Anaman-Torgbor, Lily Asare, Elizabeth A. E. Fialor.

**Writing – original draft:** Judith A. Anaman-Torgbor, Lily Asare, Bernice Osarfo, Elvis E. Tarkang.

**Writing – review & editing:** Judith A. Anaman-Torgbor.

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
