## [Decision Letter · Decision Letter 0]

24 May 2021

PONE-D-21-02324

Experiences of Women Undergoing Assisted Reproductive Technology in Ghana: A Qualitative Analysis of their Experiences

PLOS ONE

Dear Dr. Justice,

Thank you for submitting your manuscript to PLOS ONE. After careful consideration, we feel that it has merit but does not fully meet PLOS ONE’s publication criteria as it currently stands. Therefore, we invite you to submit a revised version of the manuscript that addresses the points raised during the review process.

We look forward to receiving your revised manuscript.

Kind regards,

Shah Md Atiqul Haq

Academic Editor

PLOS ONE

Additional Editor Comments:

Dear Author(s),

I would like to ask you to revise the paper by following the reviewers' comments and suggestions.

The paper has many limitations in the introduction section, organization of data and topic.

The discussion part should be strong by comparing your results with other relevant studies.

Best wishes,

Journal Requirements:

2. When reporting the results of qualitative research, we suggest consulting the COREQ guidelines: http://intqhc.oxfordjournals.org/content/19/6/349. In this case, please consider including more information on the number of interviewers, their training and characteristics; and please provide the interview guide used.

Furthermore, in your Methods section, please provide additional information about the participant recruitment method and the demographic details of your participants. Please ensure you have provided sufficient details to replicate the analyses such as: 1)  a description of any inclusion/exclusion criteria that were applied to participant recruitment and 2) a description of how participants were recruited.

3. Please ensure that you include a title page within your main document. We do appreciate that you have a title page document uploaded as a separate file, however, as per our author guidelines (http://journals.plos.org/plosone/s/submission-guidelines#loc-title-page) we do require this to be part of the manuscript file itself and not uploaded separately.

Reviewers' comments:

Reviewer's Responses to Questions

**Comments to the Author**

1. Is the manuscript technically sound, and do the data support the conclusions?

Reviewer #1: Partly

Reviewer #2: Yes

Reviewer #3: No

2. Has the statistical analysis been performed appropriately and rigorously? 

Reviewer #1: No

Reviewer #2: N/A

Reviewer #3: N/A

3. Have the authors made all data underlying the findings in their manuscript fully available?

Reviewer #1: Yes

Reviewer #2: Yes

Reviewer #3: Yes

4. Is the manuscript presented in an intelligible fashion and written in standard English?

Reviewer #1: No

Reviewer #2: Yes

Reviewer #3: Yes

5. Review Comments to the Author

Reviewer #1: 1- They should follow the correct steps to conduct a qualitative analysis. It should include reliable references upon which researchers have relied in applying the qualitative analysis.

2- they should Draft and organize the data. This could be transcribing the interview, organizing field notes from observations or ensuring all documents used in the analysis are available.

3- they should Categorize data into themes and code data elements and creating categories.

4-they should Present the collected data.

5- There are many misspellings, using capital letters in the wrong places.

6-The paper should be rearranged so that it matches the basic elements needed for the research

7-The objectives of the research and its importance should be shown more clearly. The importance of the research has not been shown well

Reviewer #2: I recommend to accept this paper. But author should address some of the issue.

1. A brief discussion (in discussio part) on, what additional knowledge this study contributing to us. (As author mentioned in Table 1)

2. Three Themes of the study should be linked properly. As I see, there is no linkage between the Themes.

3. One additional section can be added to make this study far better. i.e. If author could mention some of the existing policies taken by government, and authors view point on policy recommendation.

Reviewer #3: Thank you for the opportunity given me to give my opinion on the manuscript entitled “Experiences of Women Undergoing Assisted Reproductive Technology in Ghana: A Qualitative Analysis of their Experiences”

The manuscript is well written and well presented. It deals with an important issues. The high value placed on procreation especially in SSA showed the pain that accompanies infertility in the region.

The manuscript has several strengthen.

Below are my comments to the authors.

Introduction

Page 4:

Please add the source to the statement: “infertility is one of the reasons why some marriages end in divorce”

Please specify the date of the prevalence rates (3.5% to 16.7% in developed nations and 6.9% to 9.3% in less developed countries) of infertility specify in the introduction.

Page 5, paragraph 2: please add the source for the statements

Methodology

The abstract states that fifteen participants were invited and interviewed while in the methodology (page 7) mentions “A total of 32 women were invited to take part in the interview. However, 19 of them accepted the invitation and were successfully interviewed”. Kindly adjust

Kindly relabel, the session “Rigor and Ethical Considerations” to “Ethical considerations”

Were all interviews conducted and in English? Kindly specify and add detail if needed.

It will be good to strengthen the introduction with statistics and information about the coverage and use of ART in SSA and in Ghana, in particular. The authors include the prevalence rates of infertile which is totally fine. It will also be good if the authors include in the introduction the synthesis of the key findings of studies included in the discussion. It will provide the readers with more insight and the state of the art before the discussion.

For data analysis, I would suggest using, if possible, qualitative software data analysis. It will strengthened the robustness of the study.

The reader wants to know more about the study interviewers: were they health personnel or non-health practitioner researchers recruited outside health facility? Since the interviews took place at the health facility participants may only reported the positive experiences with health providers during the process. Kindly clarify this in the methodology and discuss, if possible, the extent to which it may affect participants’ declarations toward health practitioners providing ART services. I think this is important since it has been reported that the nature of the healthcare system play an important role ART access.

Results:

Page 14: last verbatim, second phrase, kindly remove “my” (a surplus word).

Discussion

I think the discussion session may be improved by adding more studies from sub-Saharan Africa and removing studies outside Africa if possible. Only five studies were referred to for the discussion, among which only on from SSA (South Africa). I will suggest making the discussion more clearer by including the setting (name of the countries or areas) of the studies involve in the discussion.

All of the findings presented in the current studies were in agreement with past studies according to the discussion. I will suggest the author recall clearly what is new in their study (maybe the current study is among the first in Ghanaian setting, etc….).

6. PLOS authors have the option to publish the peer review history of their article (what does this mean?). If published, this will include your full peer review and any attached files.

Reviewer #1: **Yes: **Suzan Abdel -Rahman

Reviewer #2: **Yes: **Tushar Dakua

Reviewer #3: No

---

## [Author Response · Author response to Decision Letter 0]

16 Jul 2021

Reviewer Two

Reviewer’s Remarks Author’s Response Reference Page

A brief discussion (in discussion part) on, what additional knowledge this study contributing to us. (As author mentioned in Table 1)

 The suggestion has been considered Page 19

Three Themes of the study should be linked properly. As I see, there is no linkage between the Themes.

 Statements linking the study themes have been included in the main document Pages 16 and 17

One additional section can be added to make this study far better. i.e. If author could mention some of the existing policies taken by government, and authors view point on policy recommendation

 An additional section has been included in the main document Page 22

---

## [Decision Letter · Decision Letter 1]

28 Jul 2021

Experiences of Women Undergoing Assisted Reproductive Technology in Ghana: A Qualitative Analysis of their Experiences

PONE-D-21-02324R1

Dear Dr. JONATHAN,

We’re pleased to inform you that your manuscript has been judged scientifically suitable for publication and will be formally accepted for publication once it meets all outstanding technical requirements.

Kind regards,

Shah Md Atiqul Haq

Academic Editor

PLOS ONE

Additional Editor Comments (optional):

Dear authors,

Congratulations!!!

The paper is accepted now.

Reviewers' comments:

Reviewer's Responses to Questions

**Comments to the Author**

1. If the authors have adequately addressed your comments raised in a previous round of review and you feel that this manuscript is now acceptable for publication, you may indicate that here to bypass the “Comments to the Author” section, enter your conflict of interest statement in the “Confidential to Editor” section, and submit your "Accept" recommendation.

Reviewer #2: All comments have been addressed

Reviewer #3: All comments have been addressed

2. Is the manuscript technically sound, and do the data support the conclusions?

Reviewer #2: Yes

Reviewer #3: Yes

3. Has the statistical analysis been performed appropriately and rigorously? 

Reviewer #2: Yes

Reviewer #3: N/A

4. Have the authors made all data underlying the findings in their manuscript fully available?

Reviewer #2: Yes

Reviewer #3: Yes

5. Is the manuscript presented in an intelligible fashion and written in standard English?

Reviewer #2: Yes

Reviewer #3: Yes

6. Review Comments to the Author

Reviewer #2: This manuscript should be accepted for publication. This is a very interesting topic dealing with the population of Ghana.

Reviewer #3: Comments were addressed. Thanks

Comments were addressed. Thanks

Comments were addressed. Thanks

Comments were addressed. Thanks

7. PLOS authors have the option to publish the peer review history of their article (what does this mean?). If published, this will include your full peer review and any attached files.

Reviewer #2: **Yes: **Tushar Dakua

Reviewer #3: No

---

## [Editor Report · Acceptance letter]

2 Aug 2021

PONE-D-21-02324R1 

EXPERIENCES OF WOMEN UNDERGOING ASSISTED REPRODUCTIVE TECHNOLOGY IN GHANA: A QUALITATIVE ANALYSIS OF THEIR EXPRIENCES

Dear Dr. JONATHAN:

I'm pleased to inform you that your manuscript has been deemed suitable for publication in PLOS ONE. Congratulations! Your manuscript is now with our production department. 

Kind regards, 

on behalf of

Dr. Shah Md Atiqul Haq 

Academic Editor

PLOS ONE